# A Journey in the Brain’s Clock: In Vivo Veritas?

**DOI:** 10.3390/biology12081136

**Published:** 2023-08-15

**Authors:** Alec J. Davidson, Delaney Beckner, Xavier Bonnefont

**Affiliations:** 1Neuroscience Institute, Morehouse School of Medicine, Atlanta, GA 30310, USA; dbeckner@msm.edu; 2Institut de Génomique Fonctionnelle, Université de Montpellier, CNRS, INSERM, 34094 Montpellier, France

**Keywords:** chronobiology, suprachiasmatic nucleus, calcium imaging

## Abstract

**Simple Summary:**

Most organisms living at the surface of the Earth have evolved biological clocks to cope with an environment characterized by the alternation of nights and days. For half a century, the combined efforts of scientists from various fields of expertise, including ecology, genetics, molecular biology, physiology, and neuroscience, have been engaged in deciphering the mechanisms of 24 h clocks. In mammals, the master circadian clock is located deep in the brain, which makes it difficult to reach and study in vivo, even in rodent models. This review describes historical and recent achievements, as well as future challenges and opportunities for interrogating mammalian circadian timekeeping using modern in vivo imaging.

**Abstract:**

The suprachiasmatic nuclei (SCN) of the hypothalamus contain the circadian pacemaker that coordinates mammalian rhythms in tune with the day-night cycle. Understanding the determinants of the intrinsic rhythmicity of this biological clock, its outputs, and resetting by environmental cues, has been a longstanding goal of the field. Integrated techniques of neurophysiology, including lesion studies and in vivo multi-unit electrophysiology, have been key to characterizing the rhythmic nature and outputs of the SCN in animal models. In parallel, reduced ex vivo and in vitro approaches have permitted us to unravel molecular, cellular, and multicellular mechanisms underlying the pacemaker properties of the SCN. New questions have emerged in recent years that will require combining investigation at a cell resolution within the physiological context of the living animal: What is the role of specific cell subpopulations in the SCN neural network? How do they integrate various external and internal inputs? What are the circuits involved in controlling other body rhythms? Here, we review what we have already learned about the SCN from in vivo studies, and how the recent development of new genetically encoded tools and cutting-edge imaging technology in neuroscience offers chronobiologists the opportunity to meet these challenges.

## 1. Introduction

Our brain is composed of billions of neurons and non-neuronal cells. These cells interact locally and fulfill specific tasks within specialized structures and communicate over long distances to integrate signals from remote brain areas and sensory organs. Disentangling intrinsic cell properties and their modulation by inputs from the internal and external environment is often experimentally challenging, and requires a balanced use of reductionist and holistic approaches. Circadian neurobiology seeks to investigate the biological clock that paces our physiology and behavior in tune with the natural day-night cycle. The effort to understand the mammalian brain clock has utilized a wide variety of in vitro, ex vivo, and in vivo methods to strike this balance.

This *circadian* (Latin for “around a day”) clock is physically located deep in the brain, just above the optic chiasm in the suprachiasmatic nuclei (SCN) of the hypothalamus [1,2]. Landmark studies conducted more than 50 years ago showed that endogenous 24 h biological rhythms are abolished after the destruction of the SCN [3,4]. Conversely, the transplantation of SCN neurons in place of the lesion restores behavioral rhythmicity [5], and the genetics of the donor can even drive the period of the restored rhythm [6]. These findings prompted an extensive effort to figure out how the tiny SCN, containing only around 10,000 neurons per hemisphere, generates and transfers circadian tempo and transmits it throughout the organism.

Remarkably, the SCN are intrinsically rhythmic. SCN glucose metabolism [7,8] and electrophysiological activity [9,10,11,12,13,14] are high during the day and low at night. This pattern is conserved across models, regardless of the animal’s active period (e.g., nocturnal or diurnal). This rhythmicity in vivo persists under constant darkness, even in “hypothalamic islands” containing the SCN isolated from their inputs [15]. It also persists in vitro in SCN explants and in isolated SCN neurons in culture [16,17,18,19,20,21]. This latter observation underscores the cell-autonomous nature of circadian timekeeping, which relies on a small set of clock genes engaged in transcription-translation feedback loops (TTFL) to regulate their own rhythmic expression and generate circadian rhythmicity in virtually every cell both inside and outside of the SCN [22]. In SCN neurons, this molecular clockwork controls the circadian rhythm of their electrophysiological output [21,23,24].

However, the SCN are much more than a mere collection of rhythmic cells. Several neuronal subtypes expressing specific molecular markers are localized in different regions of the SCN [25,26,27,28]. They receive different inputs and/or projects to different targets, thence fulfill different functions [25,29]. Moreover, non-neuronal cells in the SCN, such as astrocytes, regulate circadian outputs [30,31,32,33]. Altogether, SCN cells form a unique network architecture that provides essential features of the circadian pacemaker, namely its precision and robustness against perturbations. In vivo approaches to studying the SCN preserve tissue integrity and the connections with surrounding tissues. They also make it possible to assess SCN rhythms in relation to other biological parameters and environmental cues. Here, we first review what historic in vivo recording and manipulation have already told us about SCN function, mostly at a multicellular scale. We then consider the recent development of miniaturized microendoscopic devices as a game-changer in investigating the SCN cell network in freely behaving animals with high spatial and temporal resolution.

## 2. Long-Term Electrophysiology In Vivo

The SCN were discovered simultaneously by two separate groups to find that lesioning the region led to arrhythmic drinking activity (as opposed to the free-running activity of animals housed in constant darkness) [4] and arrhythmic blood corticosterone levels [3]. After the SCN were determined to be essential to daily rhythms, further fundamental experiments showed that the SCN and their component neurons expressed rhythms in vivo [8,15] and in vitro [16,17,21].

In those early days of circadian neuroscience, long-duration access to the physiology of the SCN was (and still is) technically challenging, notably the maintenance of the stability of longitudinal recordings over the course of hours and days. Recording of multi-unit activity (MUA) through extracellular electrodes aimed at the SCN proved to be a method of choice. This approach permitted direct observance of the rhythmic behavior of the rat SCN, which exhibited high activity during daytime (including subjective daytime under constant darkness conditions) and low activity at night [15]. Further, using an approach that electrically isolated the SCN within the otherwise intact brain, Inouye and colleagues showed that the SCN were the intrinsic site of the pacemaker that drives rhythms elsewhere in the brain [15]. In rats where the SCN were entirely inside the island, only the islands maintained proper circadian rhythmicity, while other parts of the brain appeared arrhythmic, indicating that the SCN synchronize the brain, if not the entire body.

MUA recordings were then extensively conducted in rats and other nocturnal species (i.e., mice and hamsters), mostly by the Meijer lab in the Netherlands. Having the circuit intact, and being able to simultaneously track longitudinally locomotor activity rhythms and sleep-wake behavior simultaneously with real-time SCN physiology, revealed that MUA rhythms are independent of food schedules [34] but rather encode environmental irradiance [35]. Indeed, this method proved very useful to our understanding of how novel photoperiods reprogram the SCN. These include day length/season [36], T-cycles [37,38], and constant light [39]. Furthermore, this approach revealed that the SCN may encode sleep states [9] and/or sleep need [40]. Most of these studies, which investigated the response to both typical and novel inputs, or required simultaneous measurement of complex behaviors or sleep, were and still are impossible to conduct with a reduced (i.e., in vitro) SCN preparation.

As powerful as this approach has been, limitations still include the lack of cell-type-specificity, and the difficulty and low throughput of identifying single units in a recording and thus understanding the role of a specific cell in the generation of the population rhythm/output. One success, however, did show that SCN subregions and individual cells can exhibit diverse and stochastic phases [41], which was verified using other in vivo methods later [42].

Recently it has become possible to target specific cell types in the SCN with electrophysiological in vivo recording [42]. The approach takes advantage of optogenetics, not as a way to manipulate the circuit, but as a means to have cell types self-identify their presence and spike waveform prior to an observational study. Such an approach may prove fruitful as a complement to single-cell imaging approaches described below, one which reports on cell output.

## 3. Microdialysis to Investigate Mechanisms of Cell–Cell Signaling In Vivo

To investigate diffusible mechanisms of intra-SCN communication, researchers in Rae Silver’s laboratory ablated wild-type and *tau* hamster SCN before restoring circadian rhythms by transplanting fetal SCN of the other genotype that had been encapsulated in a semipermeable polymeric capsule [43,44]. The capsule allowed humoral signals to diffuse from the SCN to nearby tissue but prevented neural outgrowth. Like previous experiments, the hamsters adopted circadian rhythms with a period correlating to the donor genotype upon recovery. This provided evidence that synaptic coupling is not required for entrainment as previously believed and sparked speculation about the signaling molecules that mediated this communication [43,44].

To identify which diffusible factors may be responsible for SCN synchronization, multiple groups of researchers utilized microdialysis to identify the chemical content of para-SCN interstitial fluid at various circadian times [45,46,47]. These experiments showed elevated levels of serotonin, neuropeptide Y (NPY), arginine vasopressin (AVP), vasoactive intestinal polypeptide (VIP), and gastrin-releasing protein (GRP) during the light phase that persisted in constant darkness. Release of these factors could be elicited by acute light exposure, in line with general elevated SCN activity during the light phase.

This in vivo approach again allowed for the complexity of the circuit to be maintained during the study. It further provided a means to investigate acute responses to sensory input, which is not possible in a dissected system. Weaknesses remained, however, including very low temporal and spatial resolution, and the lack of any specificity regarding which cells are being studied beyond the gross placement location of the probe. The directionality of the molecular signal cannot be determined solely via composition testing, either. Molecules present around the SCN could be delivering signals to, from, or within the SCN. Identifying the source of the signals would augment understanding, but requires other methods. The recent advent of computational techniques to infer cell–cell communication networks from single-cell transcriptomics data now provides new ways to address these issues [48,49,50].

## 4. View the Clock Ticking In Vivo

Animals that express the bioluminescent firefly protein luciferase as a reporter for clock genes, such as *Per1* and *Per2* [51,52], were developed beginning in the late 1990s. These animals opened new avenues for circadian research by enabling longitudinal monitoring of the expression of genes involved in circadian timing. Longitudinal analyses of various tissue explants from PER2::LUCIFEREASE mice revealed the SCN as the master synchronizer of self-sustained peripheral circadian oscillators, rather than the previously supposed driver of circadian rhythmicity in body organs [51,52].

Investigations in SCN slices from these reporter-gene models suggested differential function of the core and shell regions of the suprachiasmatic nucleus [53] and desynchrony of the regions after phase shift [54]. Interestingly these studies were perhaps the first to reveal heterogeneity in cellular properties such as phase [55], an intrinsic property of the network reported many times since.

A few groups successfully monitored such signals in vivo. This is possible in peripheral tissues such as salivary glands and the liver by imaging through the skin in repeatedly anesthetized animals [56], as well as in freely moving mice [57,58]. In the brain, olfactory bulb rhythms have been measured through a window over a hole in the skull using a similar repeated anesthesia methodology [59]. The SCN have also been targeted in both mice [60] and rats [61] without repeated anesthesia, using an optical fiber. However, these studies were largely confirmatory of other methods, and were limited in their scope and practicality by the noisy signal, and low temporal resolution of gene expression measurements making inferences about behavior more difficult. Furthermore, this method requires a constant infusion of the luciferin substrate for the luciferase reaction necessary for light production. These methods, to date, also do not provide a means for cellular resolution or cell-type specificity.

Recording fluorescent signals is a more typical use of in vivo optical fiber photometry. In this method, fluorescent reporters are incorporated into target cells, and, rather than an infusion of a chemical substrate, excitation light is provided through the implanted optical fiber, and the emitted light of a different wavelength is recorded by a photomultiplier tube via the same fiber. Destabilized fluorescent proteins provide short-lived reporters of circadian gene transcription. This approach was used to track the resetting of SCN gene expression in mice challenged with an abrupt light-phase shift. The authors demonstrated convincingly that circadian clock gene expression in SCN VIP+ neurons realigns faster than the daily rhythm in locomotor activity [62], confirming earlier indirect observations [63,64].

Alternatively, GCaMP reporters fluoresce in proportion to the cellular concentration of Ca^2+^, and thus can be used to visualize cell activity, both rapid and slow. Jones et al. used this approach to verify, in vivo, the key role of VIP neurons in the SCN response to light [65]. Maejima and colleagues were able to combine AVP+ cell-type-specific loss of a key neurotransmitter, GABA, with in vivo fiber photometry for Ca^2+^ [66]. This work revealed the potential for a dissociation between intracellular Ca^2+^ rhythms and behavior when cell–cell signaling is disrupted in the SCN. Another recent usage of this technique revealed the interaction between cholecystokinin-producing neurons with other SCN neuronal subtypes [67]. Utilization of this approach to in vivo recording is on the rise, and for a good reason: excellent time resolution and the capability to target specific cell populations while measuring behavior or providing environmental stimulation. However, limitations do remain in fiber photometry approaches that depict the bulk activity of an ensemble of cells. Thus, cellular diversity of function within the targeted population is completely obscured.

## 5. Imaging the SCN at a Single-Cell Resolution In Vivo

Genetic manipulations provided a means to improve temporal resolution in SCN recording and a means to focus on specific cell types. Advances in imaging technology now provide a means to refine spatial and cell–cell differences in SCN examination. Although head-fixed two-photon microscopy has enabled functional imaging of many neurological processes with little tissue damage, the procedure has limited tissue-penetrating capacity. Since the brain is typically accessed from the dorsal side, the ventral hypothalamic structures have generally been inappropriate targets for this method, although a trans-pharyngeal neural access procedure has recently been developed [68]. These methods also have limited applicability to naturalistic behaviors, as the animal must be fixed in a stereotaxic frame for the entirety of imaging. While the SCN are difficult to access directly using 2-photon microscopy, a microendoscope can allow for such access [69].

Microendocopy provides access to deep brain regions through a gradient index (GRIN) lens, implanted as an optical relay between the area of interest and the top of the skull. The subsequent attachment of miniature microscopes enables researchers to visualize the activity of large neural cell networks in free-moving mice. This approach provides a few advantages over previous methods. While it shares the advantages of in vivo intact study and cell-type targeting with fiber photometry, it provides cellular resolution, which can reveal unique characteristics such as stochasticity of rhythms, a characteristic previously described in vitro but that cannot be seen with population measures.

Recently, microendoscopic imaging of GCaMP indicators was used to track cell Ca^2+^ dynamics in the mouse SCN in vivo [42,70]. Ca^2+^ levels were high during daytime and low during nighttime in a large majority of SCN cells that all together gave rise to a conspicuous 24 h rhythm under both a regular light-dark cycle or constant darkness conditions, in line with earlier in vitro observations in slices and in vivo fiber photometry studies [65,71,72,73,74,75]. More surprisingly, when GCaMP expression was limited to AVP+ cells, only a fraction of these specific neurons were significantly rhythmic, and the subpopulation contributing to the aggregate day-night rhythm was different from one cycle to the next [42].

In addition to these slow Ca^2+^ dynamics, in vivo cell imaging gave access to fast Ca^2+^ events -either single Ca^2+^ spikes or Ca^2+^ waves resulting from bursts of action potentials- which received less attention in previous SCN studies, though they are typical of neural cells. Fast events were of relatively low amplitude and superimposed upon the large daily variations in Ca^2+^ levels [70], and their occurrence followed a daily rhythm in about half of SCN neurons. In stark contrast to earlier studies in reduced preparations in which neuronal activity is unambiguously higher during the daytime, fast Ca^2+^ events in vivo occurred evenly at any time of the light-dark cycle, with neurons exhibiting diverse phases in Ca^2+^ activity or arrhythmic activity (both non-targeted SCN cell population and AVP+ neurons). Together with earlier in vivo electrophysiological studies [12], these observations provide evidence of unforeseen timing properties of SCN cell activity in their intact environment, which were either masked or not accessible with in vitro approaches.

Beyond single-cell activity, in vivo imaging also provided insight at the network level. Indeed, the synchronous occurrence of fast Ca^2+^ events in multiple individual cells revealed extensive coordinated activity over long-distance in the SCN [70]. The quantitative and qualitative analysis of such network activity during the day-night cycle promises new insights into the SCN functioning and their integration of multiple inputs in vivo. For example, correlational analysis of AVP+ neuron pairs revealed that coherence within this cell population followed a strong rhythm, more so than single-cell variables, indicating that circuit-level properties cannot be directly inferred from individual cell activity [42].

Although these techniques are still very challenging, as the depth and small size of the SCN have historically acted as a barrier, these studies utilizing the technique revealed a more complex pattern of SCN synchrony in vivo than in vitro studies implied. Further improvements in methodology, including better genetic targeting for imaging constructs, should advance our understanding of how a robust and precise circadian pacemaker emerges from the mammalian SCN cell ensemble. In addition, the recent development of new-generation miniscopes now provides the possibility of conducting dual-color imaging. This makes it possible to track Ca^2+^ signals in two different cell populations at the same time, or to combine measurements of multiple biological parameters, such as electrical firing with voltage sensors, local blood flow with circulating fluorescently labeled dextran molecules, or neurotransmission and peptide release thanks to specific genetically encoded GPCR-based sensors. Altogether, these new tools open a new era in our way of questioning SCN physiology in vivo.

## 6. In Vivo Functional Manipulation

The future of SCN neurobiology lies not only in advanced imaging and electrophysiological techniques, which can provide cellular specificity and resolution, but in the functional manipulation of the circuit while recording behavior and/or circuit function. The development of Designer Receptors Exclusively Activated by Designer Drugs (DREADD) in the late 1990s offered further opportunities for targeting cells for acute activation. Designer drugs with high selectivity for their respective DREADDs can be applied to the brain via a transcranial cannula to activate or repress the targeted cells acutely on the order of minutes [76]. Chemogenetic techniques have been used to reveal the G-protein Ca^2+^ axis in the SCN encoding circadian time [71,77]. Optogenetic methods allow even finer temporal control of stimulation. Channelrhodopsins are channels that can be activated and deactivated with sub-second precision using light [78]. Techniques to target these channels to specific cells have been in development since the mid-2000s. The role of VIP neurons in photoperiodic response has been elucidated using optogenetic methods [79]. This technique was also instrumental in understanding how the SCN contribute to the circadian gating of osmotic homeostasis [80,81]. Currently, available microendoscopic imaging devices already provide the onboard capability to activate opsins in cells using fluorescent light of a different wavelength than that used for GCaMP imaging. The combination of these methods has the potential to reveal the causal role of single cell-type rhythms in the generation of the population output responsible for circadian behaviors and light responses. Successful recent attempts include the mapping of neural circuits involving VIP+ SCN neurons in the circadian control of aggression [82] and corticosterone [83] in mice.

## 7. Conclusions

There are obvious strengths and weaknesses to the various approaches outlined above. The early in vivo experiments examined the interface between SCN cells and the rest of the animal. This approach did a good job of illustrating the structure’s role but lacked the spatial resolution to distinguish the roles of specific cells. Ablation and transplant experiments were necessary to establish the fact that the SCN are the central clock, but do not offer any insight into how it functions.

Ex vivo studies are the opposite. By culturing tissue, researchers have been able to characterize individual cells well and determine their capability to maintain the rhythmicity of the SCN themselves but were unable to learn anything about either their activity changes upon receiving sensory input or how distal systems receive SCN outputs [84]. And the concern always remains—is the reduced prep fundamentally altered from its in vivo characteristics, either by time in vitro, by isolation from the intact circuit, or both? How can the network function normally when 95% of that network is missing? Ex vivo studies have revealed a snapshot of how the whole clock system is organized in vivo, including how inputs can alter its phasic organization [12]. While this is useful for building a bottom-up understanding of the emergent properties of the system, it does not reveal complexities of the in vivo dynamics that give rise to such reorganization of inputs, and cannot reveal the role of cell types or single cells in the generation of timing and responses that are relevant to behavioral outcomes. Also, these approaches will not be enough to predict deleterious outcomes or create biomedical interventions since they cannot provide a robust understanding of the inputs and outputs of the system and the scale of perturbations.

In vivo imaging not only combines the advantages of preceding technologies, but also does so in a way that is robust and efficient. The possibility of conducting longitudinal recordings over long time periods will provide unique access to SCN plasticity in an animal’s lifetime. For instance, this will enable us to address neural changes associated with normal and pathological aging [14,85], or under different physiological states, such as across the female reproductive cycle [86]. By making each animal its own control, researchers can leverage the statistical power of repeated-measures analysis and minimize the number of animals used, in line with Russell and Burch’s “Three Rs.”

That is not to say that in vivo imaging is without its own limitations. Unlike electrophysiological recording, it is impossible to collect data from SCN cells using in vivo imaging without augmenting such cells with genetically encoded reporters. Tools for reporter targeting, such as Cre recombinase mouse lines, cause phenotypic changes in the animals in some cases [87,88]. While such changes limit the generalizability of results, the same is true with any method using genetically modified mice. Freely-moving animals also inherently bring a risk of introducing motion artifacts into the data. The data-processing pipeline for these methods includes methods for removing motion artifacts, but care should be taken to ensure that uncompensated motion artifacts are not mistaken for physiological events. Motion artifacts can be removed by instead imaging anesthetized animals, but the benefit of a more naturalistic recording outweighs the risk of motion artifacts. Finally, it should be noted that the light used through which the cells are visualized or activate GCaMP during recordings stimulated can be a circadian stimulus in and of itself. GCaMP and some channelrhodopsins are activated by blue light near the optimal absorbance wavelength of melanopsin. Therefore, it is important to ensure that proper controls are used, and molecular tools and optical barriers are selected to minimize interference by errant light signaling, especially in non-mammalian models with extraretinal photoreactive cells.

Questions remain yet. The history of SCN research is non-linear, weaving between experiments on living creatures, examination of explanted organs, and single-cell studies. Past technologies enabled the discovery of the location, gross function, and general organization of the SCN, but mysteries surrounding their exact manner of encoding, transmitting, and modifying circadian phase and day length, integrating and responding to stimuli from a myriad of sources in a phase-dependent manner still remain. These are, in all likelihood, emergent properties of an intricately interconnected network made up of players (cell types) both known to science and yet to be discovered. Newly devised technology enables us to identify these individual players, giving us a better chance of reaching an understanding. While every scale of study yields valuable insight into clock mechanisms, the true nature of the clock can only be gleaned through real-time investigation of intact circuits with sensory input, as the fundamental purpose of the clock is to synchronize the timing of physiological processes with inputs indicating environmental conditions.

## Data Availability

Not applicable.

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
