# Peer review of "A Journey in the Brain’s Clock: In Vivo Veritas?"

_biology, 2023, doi:10.3390/biology12081136_

Round 1

Reviewer 1 Report

This is a Really lovely historical review of the progress made in the study of the SCN, with an emphasis on the tools that enabled each advance. It is both timely and interesting. It is very well written and organized. I have one major comment and several smaller suggestions.

Major comment: The review of tools and the advances each new technique permitted are clearly laid out. If possible, the authors should indicate how each of the major technical advances impacted our thinking about how the SCN works as a daily clock. I strongly urge the authors to consider what major questions remail unknown given the current state of the art, so that the reader can better understand the conceptual gains as new methods appeared, and which older ideas were rejected. As it stands, this nice paper is fundamentally  a straight “history story”  rather than a “discovery story”. 

Line 43: I suggest that “~10,000” rather than “10,000” with a reference to rats.

Line 60: Circadian rhythminstead of rhythm

Line 63-64: provide references

Line 89-92: As written there is a contradiction here. The Inouye work was correct in the long term in suggesting that tissue within the island is rhythmic. But it was wrong in the long term in suggesting that tissue outside of the island lost rhythmicity. Perhaps they only sampled limited sites, or perhaps they had limited N’s (many rats died). The laboratory of Nakamura has repeated some of this isolated island work and supports the conclusion that the original conclusions were not supported. 

Line 112: Explain in more detail how optogenetics can be used thus.

Line 121: This should be cited as work from the Silver. The work was done in Silver’s lab and Lehrman and Bittman and Gibson (all co-authors) for at least one of the papers were post-docs at the time.

Line 133: Provide a reference.

Line 114: it would be best to write out Per1 and Per2 rather than writing “the Pers”

Line 146: It would be appropriate to cite the paper by Hamada T with Honma KI (Nature 2016) as last author as this study was done in unanesthetized animals and many tissues were sampled.

Author Response

This is a Really lovely historical review of the progress made in the study of the SCN, with an emphasis on the tools that enabled each advance. It is both timely and interesting. It is very well written and organized. I have one major comment and several smaller suggestions.

We thank the reviewer for positive assessment of our manuscript. All comments have been addressed, and changes are highlighted in the attached revised version.

Major comment: The review of tools and the advances each new technique permitted are clearly laid out. If possible, the authors should indicate how each of the major technical advances impacted our thinking about how the SCN works as a daily clock. I strongly urge the authors to consider what major questions remail unknown given the current state of the art, so that the reader can better understand the conceptual gains as new methods appeared, and which older ideas were rejected. As it stands, this nice paper is fundamentally  a straight “history story”  rather than a “discovery story”. 

Thank you for this constructive comment. We have revised the last paragraph to draw the reader's attention to what remains unknown in SCN research (lines 327-336).

Line 43: I suggest that “~10,000” rather than “10,000” with a reference to rats.

Changed using "around", line 47.

Line 60: Circadian rhythminstead of rhythm

The text was changed to make the original meaning clearer, line 60.

Line 63-64: provide references

References 25 and 29 added, line 64.

Line 89-92: As written there is a contradiction here. The Inouye work was correct in the long term in suggesting that tissue within the island is rhythmic. But it was wrong in the long term in suggesting that tissue outside of the island lost rhythmicity. Perhaps they only sampled limited sites, or perhaps they had limited N’s (many rats died). The laboratory of Nakamura has repeated some of this isolated island work and supports the conclusion that the original conclusions were not supported. 

We don't agree that Nakamura's findings that SPZ contains an independent oscillator disproves Inouye's finding that substantia nigra rhythms are driven by the SCN. We made phrasing change for correctness, line 92.

Line 112: Explain in more detail how optogenetics can be used thus.

Details about how optogenetics can be used are provided in section 6, lines 266-278.

Line 121: This should be cited as work from the Silver. The work was done in Silver’s lab and Lehrman and Bittman and Gibson (all co-authors) for at least one of the papers were post-docs at the time.

Done, lines 119-120.

Line 133: Provide a reference.

References 43 and 44 added, line 127.

Line 114: it would be best to write out Per1 and Per2 rather than writing “the Pers”

Done, line 149.

Line 146: It would be appropriate to cite the paper by Hamada T with Honma KI (Nature 2016) as last author as this study was done in unanesthetized animals and many tissues were sampled.

Reference 58 added, line 162.

Reviewer 2 Report

The authors have prepared a scholarly and thoughtful review of the foundational history and emerging opportunities of in vivo circadian models, focusing on the role of the suprachiasmatic nucleus, its network organization, and its role as the pacemaker that coordinates circadian rhythms throughout the body. The review is succinct, with the perfect balance of detail and perspective.

I'll only offer these few observations as suggestions for the authors, but this reviewer would be satisfied with the paper published in it already excellent form.

1. The reader experience could be enhanced with a figure or table that might highlight the previously-intractable research questions and the novel techniques that might now be brought to bear to tackle these open questions. This is the major theme of the paper, so presenting this information in another form might augment the overall presentation.

2. The only section that felt a little weak was #3. The opening paragraph of this section presents the elegant evidence that there is a sufficient diffusible signal from the SCN to communicate rhythmicity to the rest of the brain/body, and then posits what this signal may be. The next paragraph goes to highlight the role that microdialysis has played in exploring this question, but some of the studies identified looked at signals to the SCN (5HT, NPY). Other studies mentioned looked at release related to light exposure. This is an interesting question, but is different than the question set up for this section (i.e., what signal does the SCN release to coordinate phase to peripheral tissues). There really are three interesting questions here: 1. afferent signals to the SCN, 2. intrinsic signal released and received by the SCN, and 3) efferent signals from the SCN. This section could be revised to address these important yet distinct questions. Microdialysis certainly contributes to the first two. The last question on efferent signals might require microdialysis recording at SCN targets rather than the SCN itself. Other signals not mentioned that could be include are VIP (Francl 2010) and Prokineticin2 (Morris 2021).

3. The author discuss a number of novel approaches that could be used, a few of which require delivery of light to the brain (fiber photometry, optogenetics). Given how exquisitely sensitive the circadian system is to light, particularly the blue wavelengths used in these techniques, inclusion of a cautionary note about stray light exposure and critical necessity of proper controls when using these light-producing tools would be helpful to scientists new to the circadian field. 

Author Response

The authors have prepared a scholarly and thoughtful review of the foundational history and emerging opportunities of in vivo circadian models, focusing on the role of the suprachiasmatic nucleus, its network organization, and its role as the pacemaker that coordinates circadian rhythms throughout the body. The review is succinct, with the perfect balance of detail and perspective.

I'll only offer these few observations as suggestions for the authors, but this reviewer would be satisfied with the paper published in it already excellent form.

We thank the reviewer for positive assessment of our manuscript. All comments have been addressed, and changes are highlighted in the attached revised version.

1. The reader experience could be enhanced with a figure or table that might highlight the previously-intractable research questions and the novel techniques that might now be brought to bear to tackle these open questions. This is the major theme of the paper, so presenting this information in another form might augment the overall presentation.

Thank you for this suggestion.

A table would somewhat enhance reader experience, but to build a table that effectively does so would constitute a major revision that is not practical within the allotted time for minor revision (5 days). The costs of delaying the paper outweigh the benefits of the proposed table.

2. The only section that felt a little weak was #3. The opening paragraph of this section presents the elegant evidence that there is a sufficient diffusible signal from the SCN to communicate rhythmicity to the rest of the brain/body, and then posits what this signal may be. The next paragraph goes to highlight the role that microdialysis has played in exploring this question, but some of the studies identified looked at signals to the SCN (5HT, NPY). Other studies mentioned looked at release related to light exposure. This is an interesting question, but is different than the question set up for this section (i.e., what signal does the SCN release to coordinate phase to peripheral tissues). There really are three interesting questions here: 1. afferent signals to the SCN, 2. intrinsic signal released and received by the SCN, and 3) efferent signals from the SCN. This section could be revised to address these important yet distinct questions. Microdialysis certainly contributes to the first two. The last question on efferent signals might require microdialysis recording at SCN targets rather than the SCN itself. Other signals not mentioned that could be include are VIP (Francl 2010) and Prokineticin2 (Morris 2021).

This section has been extended to address the reviewer's concern, lines 132 and 139-145.

References 47 and 50 have been added as suggested.

3. The author discuss a number of novel approaches that could be used, a few of which require delivery of light to the brain (fiber photometry, optogenetics). Given how exquisitely sensitive the circadian system is to light, particularly the blue wavelengths used in these techniques, inclusion of a cautionary note about stray light exposure and critical necessity of proper controls when using these light-producing tools would be helpful to scientists new to the circadian field. 

Thank you for this important reminder. We now discuss this limitation, lines 320-326.